# HSC-T: B-ultrasound-to-elastography Translation via Hierarchical Structural Consistency Learning for Thyroid Cancer Diagnosis

Hongcheng Han[1], Zhiqiang Tian[2], Qinbo Guo[1], Jue Jiang[3], Shaoyi Du[1,*] and Juan Wang[3,*]

*Abstract*—Elastography ultrasound imaging is increasingly important in the diagnosis of thyroid cancer and other diseases, but its reliance on specialized equipment and techniques limits widespread adoption. This paper proposes a novel multimodal ultrasound diagnostic pipeline that expands the application of elastography ultrasound by translating B-ultrasound (BUS) images into elastography images (EUS). Additionally, to address the limitations of existing image-to-image translation methods, which struggle to effectively model inter-sample variations and accurately capture regional-scale structural consistency, we propose a BUS-to-EUS translation method based on hierarchical structural consistency. By incorporating domain-level, sample-level, patch-level, and pixel-level constraints, our approach guides the model in learning a more precise mapping from BUS to EUS, thereby enhancing diagnostic accuracy. Experimental results demonstrate that the proposed method significantly improves the accuracy of BUS-to-EUS translation on the MTUSI dataset and that the generated elastography images enhance nodule diagnostic accuracy compared to solely using BUS images on the STUSI and the BUSI datasets. This advancement highlights the potential for broader application of elastography in clinical practice. The code is available at https://github.com/HongchengHan/HSC-T.

*Index Terms*—Medical image translation, elastograghy ultrasound, hierarchical structural consistency, thyroid cancer

## I. INTRODUCTION

Ultrasound imaging, known for being non-invasive and cost-effective, is the primary method for thyroid examination [1]. Elastography, an advancing technique, improves the assessment of soft tissue biomechanics. As a result, combining B-ultrasound (BUS) with elastography (EUS) is increasingly emphasized in thyroid diagnostics [2], [3]. However, elastography requires high-end equipment and skilled operation, limiting its use with portable BUS-only devices and in patients or settings where applying pressure is difficult.

This work was supported by the National Natural Science Foundation of China under Grant No. 82202183 and No. 62173269.

*Corresponding author: Shaoyi Du (dushaoyi@xjtu.edu.cn) and Juan Wang (wangjuan@xjtu.edu.cn).

[1]Hongcheng Han, Qinbo Guo and Shaoyi Du are with National Key Laboratory of Human-Machine Hybrid Augmented Intelligence, National Engineering Research Center for Visual Information and Applications, and Institute of Artificial Intelligence and Robotics, Xi'an Jiaotong University, Xi'an 710049, Shaanxi, China.

[2]Zhiqiang Tian is with School of Software Engineering, Faculty of Electronic and Information Engineering, Xi'an Jiaotong University, Xi'an 710049, Shaanxi, China.

[3]Jue Jiang and Juan Wang are with Department of Ultrasound, the Second Affiliated Hospital of Xi'an Jiaotong University, Xi'an 710000, Shaanxi, China.

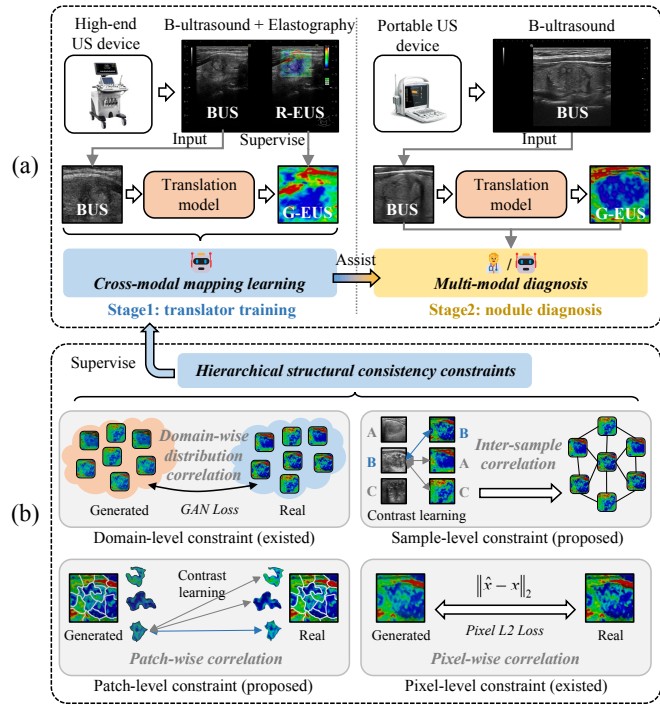

Fig. 1. Overall design of HSC-Translator. (a) Multimodal ultrasound thyroid nodule diagnostic pipeline based on BUS-to-EUS translation. (b) Illustration of hierarchical structural consistency constraints. The sample-level and patch-level constraints are proposed to address the limitations of GAN loss and pixel-wise L2 loss in modeling inter-sample relationship and capturing regional-scare structural information.

To address these issues, we propose a novel multimodal ultrasound thyroid nodule diagnostic pipeline based on B-ultrasound-to-elastography translation. As Fig.1(a) shows, it involves training a model with paired B-ultrasound (BUS) and real elastography (R-EUS) images from high-end equipment to enable it to learn the cross-modal mapping knowledge, then applying this model to obtain generated elastography images (G-EUS) in BUS-only scenarios. Combining BUS images and G-EUS images allows for improved diagnostic efficiency.

Developing an accurate and efficient translation model is crucial for implementing advanced diagnostic pipelines. Current medical image translation methods [4], [5], typically using encoder-decoder networks or diffusion models with GAN Loss and pixel-level L1 or L2 losses, have shown success in tasks like CT-to-MRI translation. However, they face significant limitations in translating BUS to EUS. First,

these methods commonly focus on reducing the distribution differences between generated and real images, neglecting inter-sample consistency and key feature differences crucial for diagnostic tasks. Second, due to the limited receptive field, pixel-level losses struggle to accurately capture the regional-scale local structural information essential for recognizing the attributes of nodules.

In response to these challenges, we propose HSC-Translator (HSC-T), a novel BUS-to-EUS translation approach that leverages multi-level losses to establish hierarchical structural consistency constraints, as shown in Fig. 1(b). Our method demonstrates superior proficiency in discerning both inter-sample consistency and distinctiveness of features through sample-wise cross-modal matching, thereby augmenting the diagnostic utility for thyroid nodule assessment. Furthermore, HSC-T enhances the representation of local structural details by adaptive patch-wise contrasting, resulting in enhanced precision in the translation from BUS to EUS. These advancements significantly elevate the diagnostic efficacy of the G-EUS in the context of thyroid nodule characterization. The main contributions of this study are as follows:

- We introduce an innovative multimodal ultrasound diagnostic workflow for thyroid nodules, which learns cross-modal mappings from paired BUS and EUS images. This approach enables the generation of virtual EUS from BUS in hardware-constrained scenarios, thus enhancing diagnostic information.
- Our novel BUS-to-EUS translation method leverages hierarchical structural consistency learning to improve inter-sample consistency, preserve local structural details, and achieve more accurate translation.
- The proposed method demonstrates superior performance compared to competing methods in BUS-to-EUS translation on the MTUSI dataset, and significantly enhances nodule diagnostic accuracy by generating G-EUS images on the STUSI and the BUSI datasets.

## II. RELATED WORK

### A. Elastography Ultrasound

Elastography ultrasound (EUS) is plays an increasingly important role in clinical diagnosis by complementing traditional B-ultrasound (BUS). Qian *et al.* [2] combined BUS, EUS, and Doppler data to create a deep-learning-based breast cancer diagnosis system, which significantly improved alignment with BI-RADS guidelines. Qin *et al.* [3] demonstrated that integrating BUS with EUS greatly enhances the accuracy of thyroid nodule diagnosis compared to using BUS alone. Zhou *et al.* [6] highlighted the importance of EUS in diagnosing lung diseases, particularly in evaluating COVID-19 pneumonia. Xu *et al.* [7] showed that EUS videos provide reliable diagnosis of intrathoracic lymph nodes, improving clinical outcomes. Additionally, Shao *et al.* [8] used shear wave elastography (S-WAVE) to extract multimodal time series features for breast cancer detection, demonstrating its ability to accurately differentiate between malignant and benign breast lesions. This growing body of work underscores the clinical significance of EUS in improving diagnostic precision across a range of conditions.

Elastography ultrasound offers clinical benefits but is limited by its reliance on high-end equipment and expert skill [9]. Since B-ultrasound (BUS) also captures some soft tissue biomechanics [10], translating BUS into elastography ultrasound (EUS) presents a promising solution to these limitations.

### B. Medical Image Modality Conversion

In response to the imperative of cost reduction, damage mitigation, and addressing the modality insufficiency in medical imaging, and driven by the advancements in computer vision's image synthesis methods [11]–[13], image-to-image translation technology is progressively being employed for modality conversion of medical images [14], [15]. For example, Kromrey *et al.* [16] achieved virtual elastography for liver fibrosis by deriving the shear modulus from diffusion-weighted MRI. Özbey *et al.* [17] enhanced multi-contrast MRI and MRI-CT translation with adversarial diffusion models, improving performance through adversarial learning. Yao *et al.* [4] utilized conditional GANs (cGANs) for converting B-ultrasound to elastography, significantly improving breast cancer diagnosis accuracy. Bharti *et al.* [18] introduced QEM-CGAN to resolve training instability, lack of diversity, and mode collapse in medical image translation using evolutionary computation and multiobjective optimization.

Current medical image translation methods focus on domain distribution or pixel-level errors but often overlook inter-sample feature relationships, crucial for BUS-to-EUS translation [19]. Pixel-level constraints, with limited receptive fields, fail to capture regional structural information needed for identifying nodule characteristics [20]. To improve BUS-to-EUS translation, enhancing both sample-level and patch-level feature learning is essential.

## III. METHODOLOGY

### A. Framework of HSC-Translator

Existing cGAN-based methods focus on minimizing overall distribution differences between generated and real data but overlook individual sample discrepancies, compromising the consistency of key personalized features. Pixel-level L1/L2 constraints, limited by their receptive field, also fail to capture regional feature consistency, leading to suboptimal outcomes and limiting the clinical use of G-EUS translated from BUS.

To address these issues, we propose HSC-Translator (HSC-T) for BUS-to-EUS translation. HSC-T incorporates hierarchical structural consistency constraints at domain, sample, patch, and pixel levels to improve both sample-wise and region-wise consistency. The framework, shown in Fig. 2, consists of two stages.

In the first stage, the translator is trained on paired BUS and EUS data, guided by hierarchical structural consistency constraints through the cross-modal matching (CMM), adaptive patch contrast (APC), and conditional adversarial (CA) modules. As shown in Fig. 2(b), the BUS input is processed

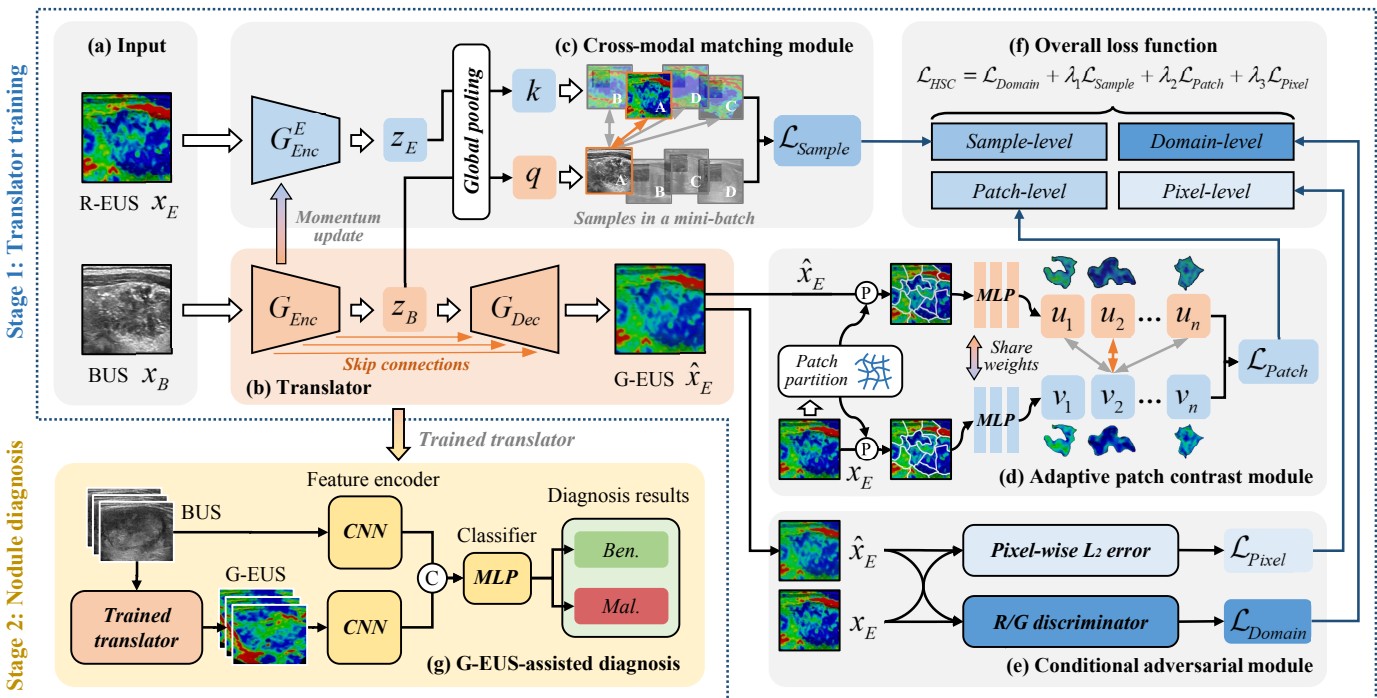

Fig. 2. Framework of HSC-Translator. (a) Input images. BUS($x_B$) is the input B-ultrasound image and R-EUS($x_E$) indicates the corresponding real elastography image. (b) Translator. An image-to-image translation network constructed using a CNN-Transformer-combined encoder $G_{Enc}$ and a transpose-convolution-based decoder $G_{Dec}$. $z_B$ is the extracted feature map and G-EUS($\hat{x}_E$) is the generated elastography image. (c) Cross-modal matching module. The extracted feature maps $z_B$ and $z_E$ are utilized to conduct sample-wise matching between B-ultrasound and elastography by calculating $\mathcal{L}_{Sample}$, modeling the inter-sample relationships. (d) Adaptive patch contrast module. It performs patch-wise contrast between $x_E$ and $\hat{x}_E$ by calculating $\mathcal{L}_{Patch}$, enhancing the capture of regional-scale features, the patch partition is obtained by pixel clustering, ⓟ denotes the process of partitioning an image into patches. (e) Conditional adversarial module. The adversarial discriminator is introduced to establish domain-level constraint $\mathcal{L}_{Domain}$ and the L2 error between $x_E$ and $\hat{x}_E$ is used to establish pixel-level constraint $\mathcal{L}_{Pixel}$. (f) Overall loss function. The hierarchical structural consistency loss function $\mathcal{L}_{HSC}$ is the weighted sum of $\mathcal{L}_{Domain}$, $\mathcal{L}_{Sample}$, $\mathcal{L}_{Patch}$ and $\mathcal{L}_{Pixel}$. (g) G-EUS-assisted diagnosis. The trained translator is applied to generate G-EUS images from portable device BUS images to assist the nodule diagnosis, the nodule classification network is constructed by two CNN encoders and a MLP classifier, ⓒ indicates concatenation.

by an encoder-decoder translator, where the encoder generates a latent representation, and the decoder produces G-EUS. The CMM module ensures sample-level consistency (Fig. 2(c)) by minimizing the distance between paired BUS and EUS samples while maximizing it between non-corresponding pairs, enhancing the translator's ability to capture unique sample features. The APC module (Fig. 2(d)) enforces region-wise consistency between G-EUS and R-EUS by performing patch contrast based on adaptive patch partitioning from pixel clustering. The CA module (Fig. 2(e)) imposes domain- and pixel-level constraints using a discriminator and pixel-wise L2 loss. Finally, the objectives of CMM, APC and CA modules are combined into the overall loss function of HSC-T (Fig. 2(f)), optimizing BUS-to-EUS translation.

In the second stage, as Fig. 2(g) shows, the trained translator is applied to generate G-EUS from BUS without the need for paired elastography images. This synthetic G-EUS data serves as an additional input to a two-stream classification network, which consists of two CNN feature encoders (both based on ResNet-34 [21]) and an MLP classifier. The integration of G-EUS significantly enhances the diagnostic performance by providing complementary elasticity information.

### B. Sample-wise Cross-modal Matching

The cross-modal matching module is proposed to minimize the distance between each BUS sample and its corresponding EUS while maximizing the distance from non-corresponding EUS samples, thereby guiding the translator to effectively learn and distinguish the unique features of each pair of samples.

As Fig. 2(c) shows, first, the feature encoder $G_{Enc}^E$ is constructed to obtain the latent representation of the elastography, which shares an identical structure with the feature encoder $G_{Enc}$ in the translator. The feature maps of the input BUS and the EUS are calculated by $z_B = G_{Enc}(x_B)$ and $z_E = G_{Enc}^E(x_E)$. Then, they are converted to the query vector $q$ and the key vector $k$ by global pooling operation, respectively. Sample-wise cross-modal matching is achieved by the contrast calculating between $q$ and $k$. Due to the large number of samples in the training dataset, the contrast calculation is conducted within a mini-batch to ensure computational efficiency. Sample-wise structural consistency loss $\mathcal{L}_{Sample}$ is calculated as

$$\mathcal{L}_{Sample} = -log\frac{exp(q \cdot k_+/\tau)}{\sum_{i=1}^{N} exp(q \cdot k_i/\tau)}, \quad (1)$$

where $N$ indicates the number of samples in the mini-batch, $q$ means the feature vector of the input BUS, $k_i$ indicates the feature vector of the $i$-th EUS in the mini-batch, and the $k_+$ represents the feature vector of the EUS that is associated with the input BUS. A scaling coefficient $\tau$ is introduced to normalize the value of $q \cdot k$ within an appropriate range, which is set to 0.05.

To address the challenge of maintaining consistent feature extraction during end-to-end training, where the separate parameter updates of encoder $G_{Enc}^E$ and $G_{Enc}$ through gradient backpropagation can lead to optimization instability and increased computational overhead, momentum update [22] is employed for optimizing the parameters of the two encoders. The parameters of $G_{Enc}^E$ and $G_{Enc}$ at the $t$-th step are calculated by

$$\theta_t = \theta_{t-1} - \eta_t \frac{\partial \mathcal{L}}{\partial \theta}, \tag{2}$$

$$\theta_t^E = m\theta_{t-1}^E + (1-m)\theta_t, \tag{3}$$

where $\theta_t$ and $\theta_t^E$ are the parameters of $G_{Enc}$ and $G_{Enc}^E$, separately. $\eta_t$ is the learning rate. $\mathcal{L}$ indicates the overall loss and $m$ is the momentum coefficient, which is set to 0.99.

Sample-wise cross-modal matching establishes the inter-sample correlations between different samples, thereby enhancing the model's attention to salient sample-specific features and improving the accuracy and clinical applicability of the elastography generated from B-ultrasound images.

### C. Adaptive Patch-wise Contrast

Pixel-level L1 and L2 losses in cGANs evaluate errors independently for each pixel, missing inter-pixel relationships and often failing to capture regional-scale features needed for accurate BUS-to-EUS translation. To address this, we propose an adaptive patch contrast module that improves inter-pixel relationship modeling, enhancing structural consistency between G-EUS and R-EUS.

As Fig. 2(d) shows, first, pixel clustering based on simple linear iterative clustering (SLIC) [23] is performed to the R-EUS $x_E$ to obtain the adaptive patch partition $\mathbb{P}(x_E) = \{p_1, p_2, ..., p_n\}$, the number of patches $n$ is set to 16. Second, G-EUS and R-EUS are divided into patches according to the partition $\mathbb{P}(x_E)$. Compared to the square divided patches, the patches obtained through adaptive pixel clustering not only consider the positional relationship but also incorporate pixel correlation, thereby augmenting the semantic and perceptual significance when assessing local structural consistency between G-EUS and R-EUS. Then, each patch of G-EUS and R-EUS is flattened and resized to a uniform size. Subsequently, the flattened patches of G-EUS and R-EUS are fed into two multi-layer perceptrons (MLP) respectively, to obtain the latent representations for patch-wise contrast learning, the patch-level loss is calculated by

$$\mathcal{L}_{Patch} = -log \frac{exp(u \cdot v_+/\tau)}{\sum_{i=1}^{n} exp(u \cdot v_i/\tau)}, \tag{4}$$

where $n$ indicates the number of patches, $u$ represents the feature vector of one of the patches in G-EUS, $v_i$ indicates the feature vector of the $i$-th patch in R-EUS, and $v_+$ represents the feature vector of the patch in R-EUS that is located at the same position as $u$. $\tau$ is the scaling coefficient introduced to normalize the value of $u \cdot v$ within an appropriate range, which is set to 0.05.

The adaptive patch-wise contrast not only enhances cross-pixel correlation through pixel-clustering but also guides the model in extracting personalized structural information from each patch of EUS through patch-wise contrast learning, thereby improving the preservation of patch-level consistency, which is essential for accurate BUS-to-EUS translation.

### D. Conditional Adversarial Module

To establish domain-level constraints and pixel-level constraints, we introduce the conditional adversarial module, as Fig.2(e) shows. The R/G discriminator incorporates a lightweight CNN classifier designed to accurately differentiate between R-EUS images and G-EUS images, performing adversarial learning with the translator via a gradient reversal layer. It supervises the translation model to generate EUS images that are indistinguishable from real data, ensuring the domain-wise distribution consistency of G-EUS images and R-EUS images in terms of color, texture, structure, etc. The domain-level loss is calculated by

$$\mathcal{L}_{Domain} = \mathbb{E}_{x_E \sim P_{REUS}}[logD(x_E)] + \\ \mathbb{E}_{x_B \sim P_{BUS}}[log(1 - D(G(x_B)))], \tag{5}$$

where $D$ indicates the domain discriminator, and $G$ represents the translator, thereby $\hat{x}_E = G(x_B)$.

Subsequently, for preserving pixel-level structural information, $L2$ loss to narrow the pixel-wise Euclidean distance between G-EUS images and the corresponding R-EUS images, The pixel-level loss is calculated by

$$\mathcal{L}_{Pixel} = \frac{1}{h \times w} \|\hat{x}_E - x_E\|_2, \tag{6}$$

where $h$ and $w$ represent the height and the width of the image.

### E. Hierarchical Structural Consistency Loss

The proposed HSC-T supervises the translator's learning process through hierarchical structural consistency constraints, with the final optimization target depicted in Fig. 2(f). Domain-level loss $\mathcal{L}_{Domain}$, sample-level loss $\mathcal{L}_{Sample}$, patch-level loss $\mathcal{L}_{Patch}$, and pixel-level loss $\mathcal{L}_{Pixel}$ are established using the CA module, the CMM module and the APC module, respectively. To integrate the structural consistency constraints across all four levels effectively, the overall loss function $\mathcal{L}_{HSC}$ is defined as a weighted sum of $\mathcal{L}_{Domain}$, $\mathcal{L}_{Sample}$, $\mathcal{L}_{Patch}$ and $\mathcal{L}_{Pixel}$, expressed as

$$\mathcal{L}_{HSC} = \mathcal{L}_{Domain} + \lambda_1 \mathcal{L}_{Sample} + \lambda_2 \mathcal{L}_{Patch} + \lambda_3 \mathcal{L}_{Pixel}, \tag{7}$$

where $\lambda_i(i=1,2,3)$ are hyper-parameters used to harmonize the influence of four components on optimization, they are set to $\lambda_1 = 0.4$, $\lambda_2 = 0.8$, $\lambda_3 = 0.4$.

TABLE I
INFORMATION OF THE MTUSI, THE STUSI AND THE BUSI DATASETS.

| Dataset | Device | Age | Number of samples | | |
|---|---|---|---|---|---|
| | | | Benign | Malignant | Overall |
| MTUSI(ours) | XBH-1 | 14-88 | 1372 | 1335 | 2707 |
| STUSI(ours) | XBH-2 | 24-79 | 505 | 387 | 892 |
| BUSI [24] | BYH-1 | 25-75 | 487 | 210 | 697 |

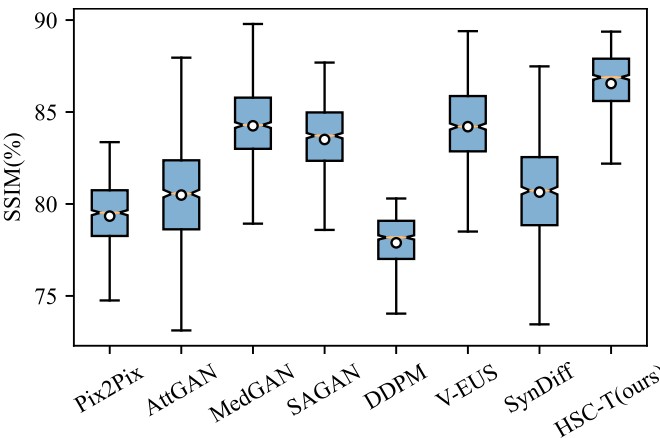

Fig. 3. Distribution of SSIM for each sample on the MTUSI dataset under different methods. The circle mark represents the mean value, the median (the horizontal line inside the box) indicates the central tendency of the data, the boundaries of the box represent the first quartile (Q1) and the third quartile (Q3), which denote the interquartile range (IQR), the whiskers extend to the smallest and largest values.

## IV. EXPERIMENTS

### A. Experimental Setup

*1) Data Preparation:* As Table I shows, to evaluate BUS-to-EUS translation performance, the MTUSI dataset was curated with 2707 paired BUS and EUS images from 2707 patients, including 1372 benign and 1335 malignant nodules.

Additionally, two other datasets were used to assess the impact of G-EUS on diagnostic accuracy. The STUSI dataset contains 824 BUS images (509 benign, 315 malignant) from devices without EUS capability. The BUSI dataset includes 780 breast nodule BUS images, comprising 133 normal tissues, 487 benign nodules, and 210 malignant ones.

Malignant cases in the MTUSI and the STUSI include thyroid papillary carcinoma, micro-papillary carcinoma, and micro-follicular papillary carcinoma, some with calcification. Benign cases include cystic degeneration, thyroid nodules, and follicular tumors, with a portion also exhibiting calcification.

*2) Implementation Details:* The datasets were split into training, validation, and testing sets (5:2:3 ratio), with consistent proportions of benign and malignant samples. Three-fold cross-validation was performed to ensure model stability, using different partitioning strategies for each experiment. In the image translation experiments, the sizes of input BUS images and output EUS images were set to $256 \times 256$. In the nodule diagnosis experiments, the sizes of BUS images and G-EUS images were set to $224 \times 224$. RMSProp optimizer is used for optimization, the initial learning rate was set to 0.001, the momentum was set to 0.95, and the weight decay was set to 1e-5. Random rotation, random cropping, and vertical and horizontal flipping were performed on the input B-ultrasound images and the corresponding ground truth elastography images for data augmentation in training.

*3) Evaluation Metrics:* For image translation evaluation, two full-reference metrics are used: structural similarity index (SSIM) and peak signal-to-noise ratio (PSNR). SSIM assesses image similarity in terms of luminance, contrast, and structure, simulating human visual perception. PSNR measures image quality by comparing the signal-to-noise ratio between real and generated EUS images. Nodule diagnosis performance is evaluated using accuracy (*Acc.*) and $F1$ score. Higher *Acc.* indicates better overall classification, while $F1$ provides a balance between sensitivity and precision.

### B. Results on BUS-to-EUS Translation

The proposed method is compared with recent influential work in the field of image-to-image translation and medical image modal conversion, including GAN-based methods [4], [11], [12], [14], [25] and also diffusion model-based methods [13], [17]. The quantitative results of various methods in the B-ultrasound-to-elastography translation task on the MTUSI dataset are presented in Table II and Fig. 3.

The results in Table II are presented in the format of $mean \pm std.$, which represents the mean value and standard deviation of the results obtained through cross-validation. The HSC-Translator outperforms existing methods in SSIM and PSNR metrics for both benign and malignant nodules, highlighting its effectiveness in improving BUS-to-EUS translation accuracy. Diffusion-based methods, DDPM [13] and SynDiff [17], perform relatively weakly, indicating the limitations of diffusion models in BUS-to-EUS mapping. The V-EUS [4] method ranks second, showing strong performance and suggesting that models for breast and thyroid ultrasound images can generalize across modality conversion tasks.

Additionally, Fig. 3 shows the distribution of SSIM indicators for each sample on the MTUSI dataset under different BUS-to-EUS translation methods. The proposed approach demonstrates superior performance and stability in the BUS-to-EUS translation task, as evidenced by its higher mean and median SSIM values, along with a narrower range between samples.

The qualitative results of a few samples are visualized and presented in Fig. 4. Compared to the competitive methods, the EUS generated by the proposed approach better preserves the overall structural information of the input B-ultrasound image and exhibits local structural details that closely resemble those of real EUS. It demonstrates that the proposed approach effectively enhances the model's capacity to acquire more robust structural consistency information, leading to improved accuracy in BUS-to-EUS translation.

TABLE II
QUANTITATIVE RESULTS OF B-ULTRASOUND-TO-ELASTOGRAPHY TRANSLATION THROUGH DIFFERENT METHODS ON THE MTUSI DATASET.

| Method | Benign | | Malignant | | Overall | |
|---|---|---|---|---|---|---|
| | $SSIM(\%) \uparrow$ | $PSNR \uparrow$ | $SSIM(\%) \uparrow$ | $PSNR \uparrow$ | $SSIM(\%) \uparrow$ | $PSNR \uparrow$ |
| Pix2Pix [11] | 79.46±4.15 | 22.54±1.39 | 79.21±4.07 | 23.75±1.13 | 79.34±4.11 | 23.35±1.12 |
| AttGAN [25] | 79.88±2.64 | 23.76±0.65 | 81.11±2.72 | 23.89±0.65 | 80.48±2.68 | 23.82±0.65 |
| MedGAN [14] | 84.24±4.57 | 25.88±1.34 | 84.17±4.66 | 25.64±1.35 | 84.20±4.61 | 25.76±1.35 |
| SAGAN [12] | 82.40±4.56 | 24.34±1.23 | 84.65±4.51 | 24.83±1.23 | 83.51±4.54 | 24.58±1.23 |
| DDPM [13] | 76.94±3.11 | 23.92±0.90 | 78.86±3.14 | 23.86±0.91 | 77.89±3.12 | 23.89±0.91 |
| V-EUS [4] | 84.85±3.02 | 25.41±0.63 | 83.64±3.03 | 26.21±0.62 | 84.25±3.03 | 25.81±0.63 |
| SynDiff [17] | 79.70±3.96 | 23.93±1.12 | 81.62±4.02 | 23.82±1.13 | 80.64±3.99 | 23.87±1.12 |
| **HSC-T(ours)** | **86.88±2.75** | **26.22±0.74** | **86.21±2.79** | **26.86±0.76** | **86.55±2.77** | **26.53±0.75** |

| Source | Reference | | HSC-T(ours) | SynDiff | V-EUS | DDPM | SAGAN | MedGAN | AttGAN | Pix2Pix |

Fig. 4. Visualization of the EUS images translated from BUS through variant methods. The first left column displays the source BUS images, while the second column exhibits the corresponding reference R-EUS images. Subsequent columns present the EUS images obtained through various translation methods.

TABLE III
RESULTS OF NODULE DIAGNOSIS BASED ON TWO-STREAM NETWORK USING BUS AND THE G-EUS GENERATED BY DIFFERENT TRANSLATION METHODS ON THE MTUSI, THE STUSI AND THE BUSI DATASETS.

| Method | MTUSI → MTUSI | | MTUSI → STUSI* | | STUSI → STUSI | | BUSI → BUSI | |
|---|---|---|---|---|---|---|---|---|
| | $Acc.(\%) \uparrow$ | $F1(\%) \uparrow$ | $Acc.(\%) \uparrow$ | $F1(\%) \uparrow$ | $Acc.(\%) \uparrow$ | $F1(\%) \uparrow$ | $Acc.(\%) \uparrow$ | $F1(\%) \uparrow$ |
| w/o EUS | 75.0±1.9 | 74.8±1.9 | 71.3±2.0 | 69.8±1.8 | 72.8±2.8 | 68.7±2.2 | 88.5±0.9 | 81.3±1.1 |
| w/ R-EUS | **89.8±2.4** | **89.7±2.3** | - | - | - | - | - | - |
| Pix2Pix [11] | 77.2±2.2 | 76.3±2.2 | 72.4±2.6 | 69.4±2.4 | 77.2±3.0 | 75.5±3.1 | 90.4±0.9 | 85.2±1.3 |
| AttGAN [25] | 77.4±2.1 | 76.9±2.2 | 76.9±3.0 | 74.6±3.2 | 79.1±2.5 | 75.2±2.7 | 91.4±1.5 | 86.6±1.4 |
| MedGAN [14] | 84.3±2.5 | 83.7±2.5 | 82.1±2.7 | 79.8±2.9 | 83.6±1.8 | 81.4±2.0 | 93.3±1.3 | 89.4±1.6 |
| SAGAN [12] | 81.6±1.9 | 81.1±1.9 | 81.0±2.2 | 78.1±2.3 | 81.3±2.4 | 78.4±2.4 | 93.3±2.0 | 89.1±1.5 |
| DDPM [13] | 79.9±2.0 | 79.3±2.1 | 73.1±2.4 | 70.5±2.4 | 75.5±2.6 | 70.7±3.0 | 89.5±1.7 | 83.8±1.6 |
| V-EUS [4] | 85.6±1.9 | 85.2±2.1 | 81.7±2.0 | 80.2±1.9 | 84.7±1.9 | 82.8±2.2 | 94.7±1.2 | 91.5±1.6 |
| SynDiff [17] | 78.9±2.4 | 78.4±2.4 | 74.3±2.1 | 69.9±2.2 | 77.6±2.3 | 73.9±2.8 | 89.0±1.2 | 83.0±1.3 |
| **HSC-T(ours)** | 87.7±2.0 | 87.4±2.0 | **85.1±2.3** | **83.2±2.2** | **86.6±1.9** | **84.7±2.0** | **95.2±1.1** | **92.4±0.9** |

* X → Y indicates training on the training set of dataset X and evaluating on the testing set of dataset Y. Therefore, MTUSI → STUSI represents a group of cross-dataset experiments.

## C. Results on G-EUS-assisted Nodule Diagnosis

Nodule diagnosis experiments were conducted to assess the impact of the generated EUS on diagnostic accuracy. The diagnostic network was trained with BUS and the G-EUS generated by different methods. To control for the network's structure, a baseline (no EUS) was tested by using BUS images in both branches. The another control (with R-EUS) compared the diagnostic efficacy of real versus generated EUS. Additionally, the cross-dataset experiments were performed to evaluate generalizability by training on the MTUSI and testing on the STUSI.

The nodule diagnosis performance of various translation methods is shown in Table.III. The introduction of G-EUS effectively enhances the diagnostic accuracy compared to solely utilizing B-ultrasound images, with the degree of improvement being positively correlated with the image translation effect demonstrated in Table II. HSC-T yields a 13.2%↑ increase in $Acc.$ and a 16.0%↑ enhancement in $F1$ score on the STUSI dataset, while achieving a 6.7%↑ improvement in $Acc.$ and an 11.1%↑ boost in $F1$ score on the BUSI dataset, which outperforms the other methods. Additionally, the G-EUS generated by the proposed HSC-T exhibits a substantial enhancement in diagnostic performance that is comparable

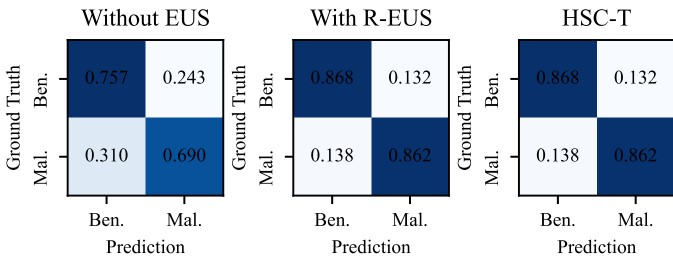

Fig. 5. Confusion matrices of the nodule diagnosis results on the MTUSI dataset when training the diagnosis network without EUS, with R-EUS and with the G-EUS generated by HSC-T.

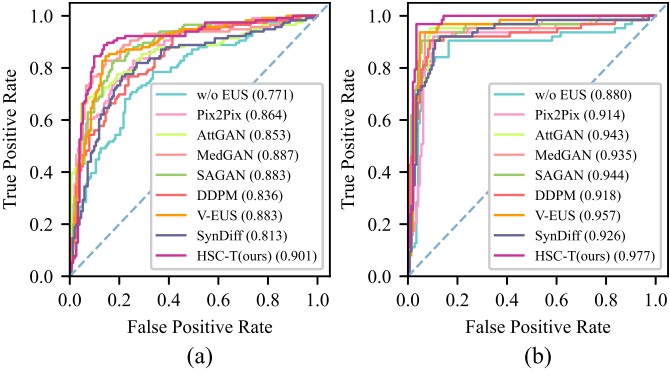

Fig. 6. RoC curves of G-EUS-assisted nodule diagnosis through various BUS-to-EUS translation methods on the STUSI and the BUSI datasets. (a) Results on the STUSI dataset. (b) Results on the BUSI dataset. The value in parentheses in the legend represents the area under the curve (AUC).

to the contribution achieved through direct utilization of R-EUS for diagnosis and exhibits significant generalizability in the experiments across the MTUSI and the STUSI datasets. Through $t$-tests conducted in four experimental groups, the p-values of the accuracy of the diagnostic model based on HSC-T compared to the diagnostic model without the use of EUS are 0.0013, 0.0014, 0.0021, 0.0012, respectively, which means the improvement in diagnostic performance brought by the G-EUS generated by HSC-T is statistically significant. Moreover, as depicted in Fig. 5, the confusion matrices further validate the potential of HSC-T in addressing the issue of missing modality within clinical practice.

Furthermore, the receiver operating characteristic (RoC) curves of G-EUS-assisted nodule diagnosis through various BUS-to-EUS translation methods are shown in Fig. 6, HSC-T achieves the highest AUC. The above findings suggest that the G-EUS generated by the proposed method can significantly enhance the diagnostic efficacy of single-mode ultrasound equipment for thyroid and breast nodules.

### D. Ablation Analysis

To investigate the impact of the proposed CMM module and APC module, ablation analysis was conducted on the MTUSI dataset by individually removing the CMM module and APC module from the model. The results are presented in Table IV, where the check mark (✓) indicates that the corresponding

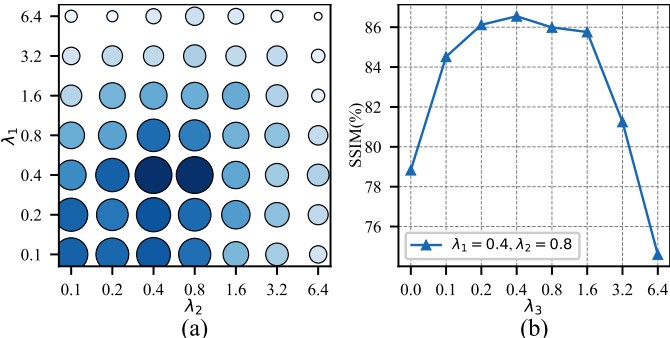

Fig. 7. Discussion on loss bias coefficients. (a) SSIM of the model with different $\lambda_1$ and $\lambda_2$ in BUS-to-EUS translation on the MTUSI dataset when $\lambda_3 = 0.4$. The size of each circle represents the SSIM performance of the model under the corresponding coefficients, with larger sizes and darker colors indicating higher SSIM values. (b) The impact of $\lambda_3$ on the SSIM performance when $\lambda_1 = 0.4$ and $\lambda_2 = 0.8$.

TABLE IV
ABLATION ANALYSIS RESULTS OF EACH MODULE IN THE PROPOSED
METHOD ON THE MTUSI DATASET.

| Module | | | Overall translation performance | |
|---|---|---|---|---|
| CA | CMM | APC | $SSIM(\%) \uparrow$ | $PSNR \uparrow$ |
| | | | 78.48±2.41 | 21.93±0.77 |
| ✓ | | | 82.12±2.90 | 24.20±0.94 |
| ✓ | ✓ | | 84.92±3.27 | 26.05±1.15 |
| ✓ | | ✓ | 84.48±2.38 | 25.73±0.65 |
| | ✓ | ✓ | 82.02±2.30 | 24.33±0.79 |
| ✓ | ✓ | ✓ | **86.55±2.77** | **26.53±0.75** |

module is included in the model. Compared to the baseline, the inclusion of the CMM module resulted in an improvement of 2.80%↑ in $SSIM$ and 1.85↑ in $PSNR$. Similarly, the inclusion of the APC module led to an enhancement of 2.36%↑ in $SSIM$ and 1.53↑ in $PSNR$. Notably, when both modules were combined, there was a significant increase of 4.43%↑ on $SSIM$ and 2.33↑ on $PSNR$. These findings demonstrate that both the CMM module and the APC module effectively improve translation from BUS images to EUS images.

Moreover, we also attempted to eliminate the CA module. However, this led to a significant decline in performance. This observation represents the effectiveness of using cGAN as the baseline for BUS-to-EUS translation. Based on the learning paradigm of cGAN, the proposed CMM and APC modules enhance the performance of image-to-image translation.

### E. Discussion on Loss Bias Coefficients

The model's bias towards sample-level, patch-level, and pixel-level consistency is controlled by $\lambda_1$, $\lambda_2$, and $\lambda_3$ in the loss function $\mathcal{L}_{HSC}$. To evaluate the effect of the proposed CMM module, APC module and CA module, these parameters were varied during training to examine their impact on model performance on BUS-to-EUS translation.

The results for 49 (7×7) different combinations of $\lambda_1$ and $\lambda_2$ are displayed in Fig. 7(a). Initially, increasing these values improves the structural similarity index (SSIM), but excessively large values lead to performance degradation. A

significant discrepancy between $\lambda_1$ and $\lambda_2$ also causes a noticeable drop in SSIM, showing that focusing too much on either sample-level or patch-level consistency negatively affects translation. Additionally, when both parameters are too high, domain-level and pixel-level consistency are neglected, further reducing performance. The optimal performance is observed when $\lambda_1 = 0.4$ and $\lambda_2 = 0.8$, indicating that this combination strikes the best balance between sample- and patch-level consistency for BUS-to-EUS translation.

To further explore the role of pixel-level constraints, Fig. 7(b) presents the impact of varying $\lambda_3$, which governs pixel-level consistency while fixing $\lambda_1 = 0.4$ and $\lambda_2 = 0.8$. SSIM drops significantly when $\lambda_3$ is either too small or too large. A small $\lambda_3$ weakens the pixel-level constraint, while a large one causes overfitting by prioritizing pixel-level consistency over other levels. The optimal setting for $\lambda_3$ is 0.4.

## V. Conclusion

In this study, we propose a novel approach for translating B-ultrasound (BUS) images into elastography (EUS) images to assist in the diagnosis of thyroid nodules. Our experimental results demonstrate that the proposed HSC-Translator effectively enhances the learning of structural consistency at different levels between BUS and EUS images. This leads to a significant improvement in the performance of BUS-to-EUS translation compared to existing methods. Additionally, when applied to portable single-modal ultrasound devices, the generated elastography (G-EUS) images produced by the HSC-Translator provide more biomechanical properties information and enhance the accuracy of thyroid cancer diagnosis compared to using BUS data alone. The ability to generate high-quality G-EUS images from BUS using our method shows promise for improving diagnostic capabilities in resource-limited settings, where access to high-end ultrasound equipment may be restricted.

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
