# OpenReview forum: "HSC-T: B-ultrasound-to-elastography Translation via Hierarchical Structural Consistency Learning for Thyroid Cancer Diagnosis"
_IEEE.org/EMBS/BHI/2024/Conference — IEEE BHI'24_

### Official Review · Reviewer_egth · 2024-07-28
**HSC-T: B-ultrasound-to-elastography Translation via Hierarchical Structural Consistency Learning for Thyroid Cancer Diagnosis**

**Overall Rating:** 7
**Confidence:** 4

**Other Quality Metrics:**

(a) Clarity of writing: excellent
(b) Clinical Significance: great
(c) Methodological Novelty: good
(d) Experiments and Results: great

**Questions For The Authors:**

none

**Strengths:**

The paper is well-written and easy to follow. The motivation for including different loss functions—such as domain-level loss, sample-level loss, patch-level loss, and pixel-level loss—is well articulated. Thorough comparison evaluations were performed, and the proposed method achieved better performance than many other state-of-the-art image translation approaches.

**Summary Of The Paper:**

This work develops an image synthesis model to translate B-mode ultrasound (BUS) to elastography ultrasound (EUS) by designing a sophisticated loss function that considers matching at different levels, including domain-level, sample-level, patch-level, and pixel-level. The authors also demonstrate the added value of translating BUS to EUS for thyroid lesion classification.

**Weaknesses:**

If the ultimate goal is to improve diagnostic accuracy, it is unclear why an intermediate step of translating BUS to EUS is necessary. Since EUS is synthesized from BUS, it supposedly will not generate more information than the original BUS images. Although the authors demonstrated the value of including synthesized EUS in lesion diagnosis, it is unclear whether the model using only BUS images is truly optimized. Some clarification on the need for EUS translation is desired.

Additionally, statistical tests are needed to demonstrate the value of the proposed method. Please clearly describe the disease when discussing the datasets.

---

### Official Review · Reviewer_2oy8 · 2024-08-10
**The manuscript presents a novel model for B-ultrasound-to-elastographyto translation, and improves the thyroid cancer diagnostic accuracy.  The method is novel, the problem to be solved is important, and the model validation analysis is sufficient.**

**Overall Rating:** 7
**Confidence:** 3

**Other Quality Metrics:**

(a) Clarity of writing: great (b) Clinical Significance: great (c) Methodological Novelty: excellent (d) Experiments and Results: excellent

**Questions For The Authors:**

why not use the MTUSI dataset in the nodule diagnosis experiment?

Clarification is needed on whether the model is trained separately for each dataset or if the STUSI and BUSI datasets share the same model parameters. This information is crucial for understanding the generalizability and robustness of the model.

**Strengths:**

The concept of applying four levels of constraints is intriguing and adds depth to the model's design.
The focus on improving thyroid cancer diagnosis addresses a critical need in medical imaging and diagnostics.
The research is very detailed and verifies the method from multiple perspectives, including comparing different models, adding ablation experiments, exploring the impact of weight coefficients, etc.

**Summary Of The Paper:**

The manuscript introduces a novel model designed to translate B-ultrasound images into elastography, with the aim of enhancing diagnostic accuracy for thyroid cancer. The proposed approach integrates four level constraints: domain-level, sample-level, patch-level, and pixel-level constraints. The proposed model, using BUS-GEUS, enhanced nodule diagnostic accuracy.

**Weaknesses:**

The manuscript should consider the issue of sample imbalance in the dataset used for classification. An unbalanced dataset can significantly impact the experimental results. It would be beneficial to include an analysis of how this imbalance affects the model's performance, perhaps using a confusion matrix. If the imbalance has a notable effect, the authors may need to consider strategies to mitigate it.

The ablation study is a critical component of validating the model. The authors should also consider extending this analysis to include domain-level and pixel-level evaluations, which could provide further insights into the model's performance and the effectiveness of the different constraints. The manuscript should also discuss the impact of the loss bias coefficient, including Lamda3, in more detail. Understanding the role of this coefficient is essential for assessing the model's optimization process.

To enhance the impact and reproducibility of the study, the authors are encouraged to make the code and model publicly available. This would allow others in the field to validate and build upon their work.

A more thorough comparison with existing models and methods in the literature is necessary. This would help position the proposed model within the broader context of thyroid cancer diagnosis and demonstrate its advantages over current approaches.

---

### Official Review · Reviewer_xWZx · 2024-08-11
**HSC-T: B-ultrasound-to-elastography Translation via Hierarchical Structural Consistency Learning for Thyroid Cancer Diagnosis**

**Overall Rating:** 7
**Confidence:** 3

**Other Quality Metrics:**

(a) great
(b) great
(c) great
(d) excellent

**Questions For The Authors:**

Data are collected from three medical devices. Will the type of medical device impact the BUS-to-EUS translation performance and nodule diagnostic accuracy? How would the performance be if we used an unseen medical device?

**Strengths:**

1. This paper presents a clear framework for HSC-Translator.
2. This paper designs effective evaluation metrics and conducts comprehensive experiments.
3. Both technical novelty and clinical contribution are good.

**Summary Of The Paper:**

This paper involves training a model with paired B-ultrasound (BUS) and real elastography (R-EUS) images from high-end equipment to enable it to learn the cross-modal mapping knowledge, then applying this model to obtain generated elastography images
(G-EUS) in BUS-only scenarios.

**Weaknesses:**

Lack of details regarding the size of training samples and testing samples.

---

### Decision · Program_Chairs · 2024-09-23

Accept